# Characterization of Mesothelin Glycosylation in Pancreatic Cancer: Decreased Core Fucosylated Glycoforms in Pancreatic Cancer Patients’ Sera

**DOI:** 10.3390/biomedicines10081942

**Published:** 2022-08-10

**Authors:** Adrià Duran, Pedro E. Guerrero, Maria Rosa Ortiz, Dúnia Pérez del Campo, Ernesto Castro, Adelaida Garcia-Velasco, Esther Fort, Rafael de Llorens, Radka Saldova, Esther Llop, Rosa Peracaula

**Affiliations:** 1Biochemistry and Molecular Biology Unit, Department of Biology, University of Girona, 17003 Girona, Spain; 2Girona Biomedical Research Institute (IDIBGI), 17190 Girona, Spain; 3Pathology Department, Josep Trueta University Hospital, 17007 Girona, Spain; 4Clinic Laboratory, Josep Trueta University Hospital, 17190 Girona, Spain; 5Hepato-Biliary and Pancreatic Surgery Unit, Josep Trueta University Hospital, 17007 Girona, Spain; 6Institut Catala d’Oncologia (ICO), Josep Trueta University Hospital, 17007 Girona, Spain; 7Department of Gastrointestinal, Josep Trueta University Hospital, 17007 Girona, Spain; 8GlycoScience Group, National Institute for Bioprocessing Research and Training (NIBRT), Fosters Avenue, Mount Merrion, Blackrock, A94 X099 Dublin, Ireland; 9UCD School of Medicine, College of Health and Agricultural Science (CHAS), University College Dublin (UCD), D04 V1W8 Dublin, Ireland

**Keywords:** pancreatic cancer, biomarkers, mesothelin, *N*-glycan, lectins, core fucose, *Pholiota squarrosa* lectin

## Abstract

Currently, there are no reliable biomarkers for the diagnosis of pancreatic cancer (PaC). Glycoproteomic approaches that analyze the glycan determinants on specific glycoproteins have proven useful to develop more specific cancer biomarkers than the corresponding protein levels. In PaC, mesothelin (MSLN) is a neo-expressed glycoprotein. MSLN glycosylation has not been described and could be altered in PaC. In this work, we aimed to characterize MSLN glycans from PaC cells and serum samples to assess their potential usefulness as PaC biomarkers. First, we analyzed MSLN glycans from PaC cell lines and then we developed an enzyme-linked lectin assay to measure core fucosylated-MSLN (Cf-MSLN) glycoforms. MSLN glycans from PaC cells were analyzed by glycan sequencing and through Western blotting with lectins. All of the cell lines secreted MSLN, with its three *N*-glycosylation sites occupied by complex-type *N*-glycans, which were mainly α2,3-sialylated, core fucosylated and highly branched. The Cf-MSLN glycoforms were quantified on PaC serum samples, and compared with MSLN protein levels. The Cf-MSLN was significantly decreased in PaC patients compared to control sera, while no differences were detected by using MSLN protein levels. In conclusion, Cf-MSLN glycoforms were differently expressed in PaC, which opens the way to further investigate their usefulness as PaC biomarkers.

## 1. Introduction

Despite its low incidence, pancreatic cancer (PaC) is the third cause of death by cancer in developed countries, presenting a 5-year survival rate of only 10%, the lowest among all of the cancers [1]. This poor prognosis is due to its high aggressiveness and the lack of symptoms, so that metastasis has usually occurred at the moment of detection. In most of the cases surgical resection is not an option, and conventional therapies, such as radiotherapy and chemotherapy, remain the only choices despite their low effectivity [2]. In addition, novel approaches, such as immunotherapy or targeted therapy, have not yet been successful in PaC management [2,3]. Thus, there is an urgent need to find new tools for the early detection of pancreatic cancer.

Currently, the most used biomarker is the carbohydrate antigen 19-9 (CA19-9), corresponding to the sialyl-Lewis a (sLe^a^) antigen, primarily used in combination with the carcinoembryonic antigen (CEA) [4]. Other biomarkers, such as CA242, albumin or IGF-1, have also been proposed to improve the CA19-9 accuracy [5,6]. Unfortunately, CA19-9 is also found in other cancers, including gastrointestinal malignancies, ovarian mucinous carcinoma or lung cancer, as well as in benign pathologies, such as chronic pancreatitis (ChP) or renal failure, decreasing its specificity and making it useless for PaC diagnosis [7]. In the clinical routine, choledocholithiasis and acute cholangitis are the main benign conditions with CA19-9 elevation, and poise a challenge in the differential diagnosis with malignant pathologies [8]. In addition, CA19-9 cannot be synthesized by genotypically deficient Le^a-b-^ individuals, that account for 5–10% of general population, thus producing false negative outcomes [9]. However, CA19-9 strictly correlates with the clinical response after pancreatectomy [4], making it a good biomarker for PaC patient-tracking [10].

Many of the proteomic strategies have been unsuccessful in discovering new biomarkers specific enough for a PaC diagnosis [11]. This deficiency could be overcome with the help of glycoproteomic approaches [12]. Altered glycosylation is known to play a major role in the development and progression of tumor cells, enhancing their survival, proliferation, migration and invasive abilities [13,14,15]. In the case of PaC, the analysis of its glycome has revealed an increase in truncated *O*-glycans, *N*-glycan branching and bisecting *N*-acetylglucosamine, as well as an overexpression of fucosylated and sialylated structures, particularly those of the sialyl-Lewis family [16,17].

*N*-glycan sialylation is altered through the overexpression of sialyltransferase enzymes, such as ST3Gal III, ST3Gal IV or ST6Gal I, which increase the PaC cells’ metastatic potential [18,19,20,21]. Moreover, increased antenna fucosylation has been described in PaC patients’ sera [22], as well as on specific glycoproteins, such as α-1 acid glycoprotein [23]. Altered core fucosylation has also been observed in specific glycoproteins, such as ribonuclease 1 [24] or haptoglobin [25]. Thus, glycosylation is an important element to consider in cancer biomarkers’ research. Actually, most of the FDA-approved tumor markers are glycan antigens or glycoproteins, although their analysis is usually only executed at the protein level [26]. However, during cancer progression, the changes in glycosylation are more marked than the protein expression alterations, thus making glycans more reliable predictive biomarkers than the proteins themselves [27]. Thus, more specific cancer biomarkers could be discovered based on the identification of singular protein glycoforms [12], being those that are measurable in blood samples the best candidates to develop into novel biomarkers.

Mesothelin (MSLN) is a glycoprotein neo-expressed in pancreatic cancer that has been proposed as a diagnostic and therapeutic target in PaC [28]. It is synthesized as a 72 kDa precursor and then cleaved to produce a C-terminal GPI-anchored 40 kDa mature form [29], which can be released from the cell membrane and reach the bloodstream [30]. Mesothelin mediates cellular adhesion through binding to MUC16 [31,32], while in cancer it is involved in resistance to cell death, cell proliferation, invasive and metastatic potential, angiogenesis, apoptosis regulation and epithelial-to-mesenchymal transition [33,34,35,36,37]. The expression of MSLN in healthy tissues is restricted to the mesothelial cells of the pericardium, peritoneum and pleura, while it is found in various malignancies apart from PaC, including ovarian cancer or mesothelioma [38,39,40,41]. This limited expression in healthy tissues makes mesothelin a useful target for therapeutic strategies. In this regard, various clinical trials concerning immunotoxins, monoclonal antibodies, antibody–drug conjugates, vaccines or CAR-T cells have been conducted for PaC and other solid tumors [34,41].

The presence of MSLN in pancreatic tissues has been examined by immunohistochemistry [42,43,44,45,46]. An abundant expression has been observed in PaC tissues, even though it is unrelated to cancer aggressiveness [47]. On the other hand, no presence has been found in the normal pancreas or in benign alterations, such as ChP [48]. However, it is not clear at which moment of cancer development the MSLN expression begins: while the pancreatic intraepithelial neoplasia does not show MSLN expression [42], MSLN is observed in the intraductal papillary mucinous neoplasms [44], with a higher frequency in the invasive lesions [45,49]. When released from the cell membrane, MSLN can be detected in sera from patients with different cancers [39,40,50]. Actually, its quantification with the Mesomark^®^ assay is routinely used for the diagnosis of malignant pleural mesothelioma [51]. In addition, the analysis of the circulating mesothelin levels in blood was proposed for PaC diagnosis. However, despite the fact that MSLN was increased in PaC patients vs. healthy individuals (from 0.58 nmol/mL to 0.66 nmol/mL) [52], it showed no utility in the diagnosis of PaC, due to low sensitivity and specificity values [50,53].

MSLN presents three putative *N*-glycosylation sites (N-X-S/T), according to its aminoacidic sequence. Moreover, MSLN *N*-glycosylation has been confirmed by a molecular weight decrease in SDS-PAGE after the complete resection of its *N*-glycans [54]. Even so, its *N*-glycans structural characterization has not been performed to date, and there is no information about the carbohydrate determinants that are more frequently displayed on MSLN under different physiological conditions.

This study aims to characterize the MSLN glycoforms in PaC samples and to develop a methodology for their quantification. First, the expression of MSLN in the PaC cell lines (lysates and conditioned media) was assessed by Western blotting (WB). Then, we characterized the glycosylation pattern of MSLN from several PaC cell lines after MSLN immunopurification, both at site-specific *N*-glycan occupancy and at the structural glycan heterogeneity level, by using mass spectrometry, *N*-glycan sequencing and blotting with lectins. We tested the different lectins that were reactive to the MSLN glycans to develop an ELLA assay to quantify the specific MSLN glycoforms in PaC. PhoSL was the lectin that provided better sensitivity, limit of detection (LOD) and limit of quantification, and was used to determine the level of MSLN core fucosylation on the tissue and serum PaC and control samples.

## 2. Materials and Methods

### 2.1. Samples

A commercial recombinant human mesothelin standard (rMSLN, #3265-MS, R&D Systems, Minneapolis, MN, USA) which is produced in a murine myeloma cell line (NS0-derived) was used throughout the study as a control, both for the MSLN expression and glycosylation.

#### 2.1.1. Cell Lines

Seven PaC cell lines from the America Type Culture Collection (ATCC) and one ovarian cancer cell line (Ovcar-8) from the National Cancer Institute (NCI) were used in this study. The Capan-1, Capan-2, SW1990, BxPC-3 and AsPC-1 were maintained in DMEM supplemented with 10% FBS and 1% penicillin/streptomycin. The HPAF-II cells were maintained in EMEM supplemented with 10% FBS and 1% penicillin/streptomycin. The Panc 10.05 cells were maintained in RPMI supplemented with 15% FBS, 1% penicillin/streptomycin and 10 mg/L insulin. The Ovcar-8 cells were maintained in RPMI supplemented with 10% FBS and 1% penicillin/streptomycin. All of the cells were cultured at 37 °C in a humidified atmosphere containing 5% CO_2_. The cell growth and morphology were assessed daily using an inverted microscope.

The protein lysates were obtained from fresh cultures in a 10 mm plate. Following medium aspiration and PBS washing, 500 μL of RIPA buffer (50 mM Tris pH 7.4, 150 mM NaCl, 0.1% NP-40, 0.5% Na-deoxycholate, 0.1% SDS, 2 mM EDTA, 50 mM NaF, 1 mM PMSF, 0.2 mM Na3VO4, protease inhibitors Complete ULTRA Tablets™ (Roche Diagnostics, Mannheim, Germany) were added. After 10 min incubation at 4 °C, the plates were scratched and the lysates recovered in 1.5 mL tubes, for being passed next through a needle 25 times and incubated for 15 extra minutes at 4 °C. Finally, the lysates were centrifuged for 10 min at 14,000 g, and the supernatant recovered and stored at −20 °C until its use. Total protein quantification was performed, using the Bradford assay (Bio-Rad Laboratories, Hercules, CA, USA) following the manufacturer’s instructions.

For the study of the cell-secreted proteins, the conditioned media of the cell lines was collected. In short, the cells were washed with PBS when they reached 80% confluence. Next, the cells were grown in free-FBS medium, which was collected 48 h later and concentrated by centrifugation in 10 kDa filters (Merck Millipore, Burlington, MA, USA) previously passivated with 5% Brij-35. As for the protein lysates, the total protein concentration was measured by the Bradford assay.

#### 2.1.2. Human Samples

Human samples were provided by the Hospital Dr. Josep Trueta (Girona, Spain), following the standard operation procedures of its Ethics Committee, in accordance with the current Declaration of Helsinki, the European Regulation (EU) 2016/679 and the Spanish Organic Law 3/2018 on data protection. The ethical approval for this study was obtained from the Comitè d’Ètica d’Investigació Hospital Universitari Dr. J. Trueta (Girona, Spain), reference number 2021.005. Informed written consent was obtained from all of the participants. The cancer samples were confirmed by biopsy or image examination by the digestive and pathology units, and classified according to the Tumor Node Metastasis Classification of Malignant Tumors of the International Union Against Cancer (UICC), 8th edition. 

A small piece of the resected pancreatic tissues was immediately frozen in liquid nitrogen and kept at −80 °C until its use for protein extraction. The protein extraction was performed from 43 frozen tissues: thirty-one PaC tissues at different stages; two tissues from other gastrointestinal malignancies and ten control tissues. The controls included two healthy pancreas from autopsies and eight pancreatic non-tumor (NT) tissues adjacent to the cancer region (two from cholangiocarcinomas, two from duodenum adenocarcinomas, one from an ampullary carcinoma and three from PaC) (Table 1). The whole protein homogenates were obtained through lysis with lysing matrix beads D (MP Biomedicals, Santa Ana, CA, USA) in a FastPrep-24 Instrument (MP Biomedicals, Santa Ana, CA, USA) in RIPA buffer (50 mM Tris pH7.4, 150 mM NaCl, 0.1% NP-40, 2 mM EDTA, 50 mM NaF, 1 mM PMSF, 0.2 mM Na3VO4, protease inhibitors Complete ULTRA Tablets^TM^ (Roche Diagnostics, Mannheim, Germany). The total protein quantification was determined by the Bradford assay (Bio-Rad Laboratories, Hercules, CA, USA), following the manufacturer’s instructions.

Blood serum was collected from 31 individuals, including five healthy donors, eleven ChP patients and fifteen PaC patients at different stages (Table 2). The serum was collected following standard procedures, aliquoted and stored at −80 °C until its use. The mesothelin levels were quantified with Mesomark^®^ (Fujirebio Diagnostics, Malvern, PA, USA) by Reference Laboratory, Barcelona, Spain.

### 2.2. In Solution Mesothelin Glycosidase Digestion

The complete release of *N*-glycans from the protein samples was achieved by digestion with *N*-glycosidase F (PNGaseF) (New England Biolabs, Ipswich, MA, USA), following the manufacturer’s instructions. Briefly, 1 µL PNGaseF was added to 10 µg of total protein in the recommended conditions. The reaction took place at 37 °C overnight.

For the samples to be analyzed by mass spectrometry, the MSLN was previously reduced in 10 mM DTT for 15 min at 65 °C and then alkylated in 50 mM IAA for 30 min in the dark. Two units of PNGaseF per 10 µg protein were added and incubated overnight at 37 °C.

The sialic acid (SA) was selectively digested with the use of different sialidases. For specific α2,3 linkage digestion, streptococcus pneumoniae neuraminidase (NAN1, Agilent Technologies, Santa Clara, CA, USA) was used, and 50 mU was added to 10 µg total protein. The reaction took place in 50 mM sodium phosphate pH 6 at 37 °C overnight. The digestion of the SA (both α2,3- and α2,6-linked) was achieved through Arthobacter ureafaciens sialidase (ABS, Roche Diagnostics, Mannheim, Germany) digestion. Ten milliunits ABS per 10 µg total protein were used, and incubated in 20 mM phosphate buffer at 37 °C overnight.

### 2.3. Western Blotting (WB)

WB was performed following the standard procedures. Briefly, equal amounts of protein samples were loaded in Laemmli buffer under non-reducing conditions, unless otherwise stated, and resolved in 10% acrylamide SDS-PAGE. The proteins were transferred to PVDF membranes for 4 h at 100 V in Towbin buffer (25 mM Tris, 192 mM glycine, 20% *v*/*v* methanol, pH 8.3). The membranes were blocked for 1 h in 5% skimmed milk in TBST (10 mM Tris, 100 mM NaCl, 0.05% *v*/*v* Tween-20, pH 7.4) with shaking. After the TBST washing, the membranes were incubated overnight at 4 °C with the corresponding antibody: anti-MSLN, clone MN-1 (LifeSpan Biosciences, Seattle, WA, USA) diluted 1/1000 in TBST with 3% skimmed milk; anti-GADPH, clone 1E6D9 (Proteintech, Rosemont, IL, USA) diluted 1/10,000 in TBST 5% skimmed milk. Following the TBST washing, the membranes were then incubated for 1 h with an HRP-conjugated goat anti-mouse antibody (Merck Millipore, Burlington, MA, USA) diluted 1/10,000 in TBST with 0.5% skimmed milk. The membranes were finally developed with Immobilon™ Western Chemiluminescent HRP substrate (Merck Millipore, Burlington, MA, USA). The chemiluminescence was visualized using the imaging system Fluorchem SP (Alpha Innotech, San Leandro, CA, USA), under non-saturating conditions.

For the glycan determinants’ detection, the membranes were blocked in the TBST with 2% polyvinylpyrrolidone overnight with shaking at room temperature. They were washed in TBST, and then incubated for 2 h with different lectins (Table 3) in lectin buffer (100 mM Tris, 150 mM NaCl, 1 mM CaCl_2_, 1 mM MgCl_2_, pH 7.6). After washing with the TBST, the membranes were incubated for 1 h with HRP-conjugated streptavidin (GE Healthcare, Chicago, IL, USA), antibodies anti-FITC HRP conjugated or antibodies anti-DIG HRP conjugated (Roche Diagnostics, Mannheim, Germany) depending on the label of the used lectin. Following the TBST washing, the membranes were revealed for MSLN detection.

The membranes were reblotted, using standard stripping protocols. These were washed with TBS (10 mM Tris, 100 mM NaCl pH 7.4) before incubation with Restore Western Blot Stripping buffer (Thermo Fisher Scientific, Waltham, MA, USA) for 20 min at 37 °C. Next, the membranes were washed with TBS and TBST and developed under the WB protocol starting from the blocking step.

### 2.4. Mesothelin Immunopurification

For the MSLN immunopurification from cell lines-conditioned media, equal amounts of total protein (for WB analysis) or MSLN (for *N*-glycan sequencing analysis) were diluted in 300 µL incubation buffer (50 mM Tris pH 7.4, 150 mM NaCl, 2 mM EDTA, 1 mM PMSF, 0.2 mM Na3VO4, protease inhibitors Complete ULTRA Tablets™ (Roche Diagnostics, Mannheim, Germany)) and added to 100 µL protein G-sepharose beads (GE Healthcare, Chicago, IL, USA), previously washed in a spin-tube (Costar, Corning Inc., Corning, NY, USA). The samples were precleared for 2 h at 4 °C and were then centrifuged for 2 min at 5000× *g*. On the other side, the anti-MSLN antibody clone MN-1 (LifeSpan Biosciences, Seattle, WA, USA) was bound to the protein G magnetic beads (SureBeads™, Bio-Rad Laboratories) in the TBST, 0.001% BSA buffer for 45 min at shaking (for *N*-glycan sequencing) or to protein G-sepharose beads (GE Healthcare, Chicago, IL, USA) in TBST, 0.01% BSA for 2 h at rolling (for WB analysis). Next, the precleared fractions were added to the antibody/magnetic beads complex and were incubated for 4 h at shaking at room temperature. The samples were eluted after incubation with 40 µL 2× Laemmli buffer for 10 min at 70 °C before beads’ magnetization.

The mesothelin was immunopurified from both tissue protein lysates and blood serum samples before its ELISA analysis. First, a mouse anti-mesothelin antibody (clone MN-1, LifeSpan Biosciences, Seattle, WA, USA) was covalently bound to the magnetic beads (Dynabeads™ M-270 Epoxy, Thermo Fisher Scientific, Waltham, MA, USA), following the manufacturer’s procedures. For each sample, 50 µL of the antibody-beads complex were equilibrated twice with PBST (phosphate-buffered saline, 0.05% Tween) with 0.1% BSA. Next, 50–80 µg total protein from lysates (corresponding to 6 ng of MSLN) in 1 mL RIPA buffer or 1 mL blood serum were incubated for 1 h shaking at room temperature. The tubes were magnetized, the supernatant removed and beads washed with PBST. The MSLN was eluted in 44 µL citrate buffer 0.1 M, pH 3.1. After 2 min incubation, the tubes were magnetized, the samples collected and neutralized with 16 µL Tris 1 M pH 9. In the case of the serum samples, the eluted fraction was diluted with 900 µL RIPA buffer and incubated again for 1 h with 50 µL antibody beads-complex, previously equilibrated. The beads were washed and the MSLN finally eluted, following the same steps as before.

### 2.5. Silver Staining

The MSLN purification was verified by silver staining of the acrylamide gels. The collected fractions were loaded in Laemmli buffer and resolved by SDS-PAGE. The proteins were fixed for 1 h in 50% methanol, 12% acetic and 0.5 mL/L formaldehyde. After three 20 min washes with 50% ethanol, the gels were oxidized for 1 min with 0.2 g/L Na_2_S_2_O_3_, followed by 20 min impregnation with 2 g/L AgNO_3_ and 0.75 m L/L formaldehyde. The samples were developed for about 10 min in 60 g/L Na_2_CO_3_, 4 mg/L Na_2_S_2_O_3_, 0.5 mL/L formaldehyde. The gels were washed twice in water and development stopped with 50% methanol, 12% acetic for 10 min. The gels were finally washed in 50% methanol for 20 min.

### 2.6. N-Glycan Sequencing of MSLN Glycans

The MSLN *N*-glycans were sequenced, using the methodology described by Royle et al. [56] with slight modifications. Briefly, the immunopurified MSLN was separated on SDS-PAGE and stained with Coomassie blue. The bands corresponding to MSLN were excised, washed with 20 mM NaHCO_3_ and acetonitrile (ACN) before reduction with 100 µL 50 mM DTT for 10 min at 65 °C and alkylation with 100 µL 20 mM IAA for 30 min in the dark. Then, the gels were washed several times with acetonitrile and 20 mM NaHCO_3_. After totally drying the gel pieces in the vacuum centrifuge, 100 µL PNGaseF (New England Biolabs, Ipswich, MA, USA) diluted 1/400 in 20 mM NaHCO_3_ were added, plus 100 µL extra 15 min later. The gels were topped up with 20 mM NaHCO_3_ buffer and incubated at 37 °C overnight. The glycans were totally eluted with water and acetonitrile sonication. They were then filtered and completely dried in the vacuum centrifuge overnight. The *N*-glycans were then labelled with 2-aminobenzamide (2-AB), as described in [57]. 

The derivatized *N*-glycans were digested for 16 h at 37 °C, using several exoglycosidases from Prozyme (Santa Clara, CA, USA) unless stated otherwise, either alone or in combination: 0.5 U/mL ABS (Arthobacter ureafaciens sialidase, digests α2,3/6/8/9-SA); 5 U/mL NAN1 (Sialidase S, digests α2,3-linked *N*-acetylneuraminic acid); 1 U/mL BTG (Bovine testes β-galactosidase, digests β1,3/4 galactose); 400 U/mL AMF (Almond meal α-fucosidase, digests α1,3/4 fucose; from New England Biolabs, Ipswich, MA, USA); 1 U/mL BKF (Bovine kidney α-fucosidase, digests α1,2/3/4/6 fucose, also core fucose); 8 U/mL GUH (Streptococcus pneumoniae hexosaminidase, digests β-*N*-acetylglucosamine) and 25 U/mL CBG (Coffee bean α-galactosidase, digests α1,3/4 galactose). The enzymes were added for a total volume of 10 µL in 50 mM sodium acetate pH 5.5 buffer. After digestion, the exoglycosidases were inactivated at 65 °C for 15 min and removed through filtration in 10K microcentrifuge filtration devices (Pall Corporation, Port Washington, NY, USA).

The 2-AB labelled *N*-glycans were analyzed by ultra-performance liquid chromatography (UPLC) with fluorescence detection, on a Waters Acquity UPLC H-Class system consisting of a quaternary solvent manager, sample manager and fluorescence detector under the control of Empower3 software (Waters, Milford, MA, USA). The *N*-glycans were separated through hydrophilic interaction chromatography (HILIC) in an Acquity UPLC Glycan BEH amide column, 2.1 × 150 mm, 1.7 µM BEH particles. The temperature of the column was set at 40 °C. Solvent A was 50 mM ammonium formate pH 4.4 and solvent B was acetonitrile (ACN). A 30-min method with a linear gradient 70–53% ACN at 0.56 mL/min was used. The samples were injected in 20 µL 70% ACN. The fluorescence excitation/emission wavelengths were λ_ex_ = 330 nm and λ_em_ = 420 nm, respectively. The retention times were converted into glucose units (GU) by time-based standardization against a dextran ladder.

### 2.7. UPLC-ESI-QTof Analyses

#### 2.7.1. Intact Protein Analysis

The mass spectrometry analysis of the intact MSLN was carried out in rMSLN. The samples were prepared in a total volume of 10 µL and injected into a Waters Acquity UPLC coupled to a Waters XEvo G2 QTof under the control of MassLynx 4.1 software (Waters, Milford, MA, USA). An Acquity UPLC Protein BEH C4 column, 300 Å, 1.7 µm, 2.1 × 50 mm was used, set at 80 °C. The flow rate was 0.2 mL/min. Solvent A was H_2_O and solvent B was ACN, both with 0.1% formic acid (FA). A 25-min gradient from 30 to 45% solvent B was used for the protein separation. The mass spectrometer operated in the positive ionization mode. The capillary voltage was set at 1.5 kV, the cone voltage at 40 V and the extraction cone at 4.0 V. The source and desolvation temperatures were set at 120 °C and 600 °C, respectively. The gas flows were 20 L/h for the cone gas and 500 L/h for the desolvation gas. The scan range (500–5000 *m*/*z*) was calibrated with NaI. A lock mass correction with leucine-enkephalin was used throughout the experiments. The extracted ion chromatograms were peak-detected and the noise-reduced in both of the LC and MS domains, and then deconvoluted using the mMass software.

#### 2.7.2. Peptide Analysis

The peptide analyses were performed in both the gel and in-solution samples. The latter were dried, resuspended in NH_4_HCO_3_, DTT-reduced and IAA-alkylated before digestion with trypsin (Promega, Madison, WI, USA) at a trypsin: protein ratio of 1:10 for 16 h at 37 °C. The gel pieces were already reduced and alkylated. They were washed, dried and rehydrated with trypsin 1:50 in 50 mM NH_4_HCO_3_ for 16 h at 37 °C. The peptides were extracted from the gel pieces with three washes by sonication in 200 µL ACN:H_2_O:FA 50:50:1. The samples were dried and resuspended with 0.1% FA H_2_O to the desired concentration.

The peptides were separated and detected in the same system used for intact protein experiments, but with an ACQUITY UPLC Peptide BEH C18 column, 130 Å, 1.7 µm, 2.1 × 150 mm. Solvent A and solvent B were water 0.1% FA and ACN 0.1% FA, respectively. The flow rate was set at 0.2 mL/min, and column temperature at 40 °C. The peptides were separated in a 40-min lineal gradient from 10 to 28% solvent B. The mass spectrometer operated in the positive ionization mode. The capillary voltage was set at 3.0 kV, the cone voltage at 25 V and the extraction cone at 4.0 V. The source and desolvation temperatures were set at 120 °C and 350 °C, respectively. Gas flows were 2 L/h for the cone gas and 800 L/h for the desolvation gas. The scan range (200–2000 *m*/*z*) was calibrated with leucine-enkephalin, which was also used for the lock mass correction. The extracted ion chromatograms were peak-detected and noise-reduced in both of the LC and MS domains.

### 2.8. Enzyme-Linked Immunosorbent Assay (ELISA) to Quantify Mesothelin

A sandwich enzyme-linked immunosorbent assay (ELISA) was developed to quantify the MSLN concentration. The ELISA 96-well plates (Thermo Fisher Scientific) were coated with 300 ng anti-MSLN rabbit polyclonal antibody (LifeSpan Biosciences, Seattle, WA, USA) in 100 µL Na_2_CO_3_-NaHCO_3_ pH 9.6 buffer at 4 °C overnight. After washing with saline solution (0.9% NaCl, 0.05% Tween-20), the wells were blocked with 400 µL 1% BSA PBST (phosphate-buffered saline, 0.05% Tween) for 1 h. Then, the samples diluted in PBST were incubated in triplicate for 2 h. A calibration curve, with rMSLN ranging from 0.25 ng/mL to 10 ng/mL in PBS-T, was included. Following the washes, 100 µL anti-MSLN antibody clone MN-1 (LifeSpan Biosciences, Seattle, WA, USA) diluted 1/1000 in 1% BSA PBST were added to each well and incubated for 2 h. Next, the HRP-conjugated goat anti-mouse antibody (Merck Millipore, Burlington, MA, USA) diluted 1/3000 in 1% BSA PBST was incubated for 1 h. All of the incubations were completed in a humid chamber. The plates were developed with 100 µL/well BM blue HRP substrate (Roche Diagnostics, Mannheim, Germany), and the reaction was stopped after 10 min with 100 µL 1M H_2_SO_4_. The optical density was determined in a plate reader (Synergy 4, BioTek, Winnoski, VT, USA) at λ = 450 nm with correction at λ = 690 nm. Negative controls were the wells without a sample. The data are expressed as the mean and SD of at least two independent assays.

### 2.9. Enzyme-Linked Lectin Assay (ELLA) to Quantify Core Fucosylated Mesothelin

The MSLN core fucosylated-glycoforms were quantified, using an enzyme-linked lectin assay (ELLA). The protein G-coated ELISA plates (Thermo Fisher Scientific, Waltham, MA, USA) were washed with saline solution (0.9% NaCl, 0.05% Tween-20) and incubated with 300 ng anti-MSLN antibody clone MN-1 (LifeSpan Biosciences, Seattle, WA, USA) in 100 µL PBST for 1 h. After washing, the plates were blocked with 400 µL 2% polyvinylpyrrolidone in PBS for 1 h, followed by the incubation of samples in triplicates diluted in PBST for 2 h. A calibration curve with rMSLN ranging from 1 ng/mL to 40 ng/mL was used. Next, 100 µL of biotinylated PhoSL [55] at 1 µg/mL in lectin buffer (100 mM Tris pH 7.6, 150 mM NaCl, 1 mM CaCl_2_, 1 mM MgCl_2_) were added for 2 h. Streptavidin-HRP, conjugated from Vectastain^®^ ABC-HRP kit (Vector Laboratories, Burlingame, CA, USA), was used for signal amplification, following the manufacturer’s procedures. All of the incubations were completed in a humid chamber. The wells were revealed with 100 µL BM blue HRP substrate (Roche Diagnostics, Mannheim, Germany) for about 2 min, before stopping the reaction with 100 µL 1 M H_2_SO_4_. The optical density was determined in a plate reader (Synergy 4, BioTek, Winnoski, VT, USA) at λ = 450 nm with correction at λ = 690 nm. The negative controls were the wells without a sample. The data are expressed as the mean and SD of at least two independent assays.

### 2.10. Statistics

The statistical analyses of the obtained results were performed using Prism 9 (GraphPad Software, San Diego, CA, USA). The data from ELISA/ELLA were analyzed for normality with the Shapiro–Wilk test. After meeting this criteria, one-way ANOVA with Tukey’s post-hoc comparison was used when comparing the three groups, while unpaired *t*-test was performed for the two-groups mean comparison. For non-normally distributed data (MSLN quantification), Kruskal–Wallis and Mann–Whitney tests were performed for the same comparisons. The receiver operating characteristic (ROC) curves were analyzed for MSLN and core fucosylated-MSLN for distinguishing between the PaC and the control groups. For all of the analyses, *p* < 0.05 was considered statistically significant.

## 3. Results

### 3.1. Mesothelin Expression in Cell Lines

First, we assessed the MSLN expression in cell lines by WB. The MSLN was observed in protein lysates from all of the analyzed PaC cell lines: Capan-1; Capan-2; SW1990; BxPC-3; AsPC-1; Panc 10.05 and HPAF-II, with different intensities (Figure 1a). An ovarian cancer cell line, Ovcar-8, reported to express MSLN, and recombinant mesothelin (rMSLN) were used as the controls for MSLN expression. In addition to the mature form, the MSLN precursor (a band of about 70 kDa) could also be detected in some of the cell lines. It must be noted that mature MSLN appeared as a broad band (between 40 and 55 kDa approximately), which slightly differs in molecular weight between the cells, thus suggesting a possible range of isoforms.

MSLN can be shed from cells, despite being a GPI-anchored glycoprotein. Actually, this circulating form is the most interesting from the biomarker point of view, as it is the one potentially detected in sera. For this reason, we also analyzed the presence of MSLN in the secreted media of cell lines (Figure 1b). The observed intensity pattern was similar to that obtained for the total protein lysates, despite the fact that the precursor was obviously not detected in this case. The MSLN abundance in the culture supernatant was higher than in cell lysates when loading similar amounts of total protein. Thus, we selected the conditioned media of the cell lines with higher MSLN expression (Capan-2, AsPC-1 and HPAF-II PaC cells and Ovcar-8) for the subsequent studies on MSLN specific glycosylation. The commercial rMSLN used as a positive control was also added to the glycosylation analysis, as it is produced in a murine myeloma cell line (NS0-derived), which can also glycosylate their secreted proteins.

### 3.2. Mesothelin N-Glycosylation: Site Occupancy

The MSLN *N*-glycosylation was first checked through *N*-glycan digestion with PNGaseF. A molecular weight decrease in gel electrophoresis after PNGaseF treatment detected by WB indicated the *N*-glycans removal in the four selected cell lines protein lysates and in rMSLN (Figure 1c). The release of the *N*-glycans from rMSLN was confirmed by WB with AAL (specific lectin for Fucα3/6GlcNAc detection). A single band was detected in the non-treated rMSLN, while no signal was observed after PNGaseF digestion, confirming the removal of *N*-glycans in the digested sample (Figure 1c).

The glycosylated MSLN from cell lines lysates was observed as a single broad band, ranging from 43 to 55 kDa, which suggests a large number of isoforms. After PNGaseF digestion, a single band at around 35 kDa was detected, coinciding with the computed mass for the human MSLN polypeptide sequence (UniProt reference Q13421, 296-606aa). Regarding the rMSLN, a major band of 43 kDa and a minor band of 39 kDa were shifted to 33 kDa after the PNGaseF digestion, again the theoretical mass of the non-glycosylated rMSLN. For both the cell lines and the rMSLN, the molecular weight shift after PNGaseF digestion confirmed the presence of *N*-glycans on MSLN, and suggested a rather high glycosylation rate due to a large mobility variation.

The exact mass change on the commercial rMSLN before and after PNGaseF digestion was determined through an intact protein analysis by MS (ESI-QTof). Upon PNGaseF digestion, the glycoprotein mass shifted from 43.9 kDa to 33.3 kDa due to *N*-glycan release, confirming the results obtained by WB analyses (Figure 1c). This 10.6 kDa reduction suggested that rMSLN would carry three *N*-glycan chains.

Subsequently, we next analyzed the percentage of occupancy of the three theoretical MSLN *N*-glycosylation sites by peptide mapping (Table 4). For that, the MSLN SDS-PAGE bands from all of the samples were digested in gel with PNGaseF to remove the *N*-glycans, and then the protein was digested with trypsin. The removal of the *N*-glycans by PNGaseF causes the deamidation of asparagine (N) to aspartic acid (D) in the peptides, which implies a +0.98 Da shift in their molecular weight. The detection (or not) of this mass shift in a peptide containing a putative *N*-glycosylation site shows whether the asparagine was (or was not) glycosylated.

The three potentially *N*-glycosylated peptides (peptide 92-101, peptide 187-200 and peptide 215-230) were detected by UPLC-ESI-QTof MS in rMSLN. All three of them were only identified with a mass corresponding to the D-X-S/T sequence, thus indicating a full site occupancy for all of the *N*-glycan sites (Table 4). Concerning the MSLN from the cell lines-conditioned media, peptide 92-101 was detected in all of the samples, with full occupancy. For Capan-2, peptide 187-200 was also detected, and again fully occupied. Peptide 187-200 could not be detected in the rest of the samples, as well as peptide 215-230, probably attributable to the low MSLN amounts and the lower ionization of these peptides compared to peptide 92-101 (Table 4).

The analysis of the tryptic peptides of PNGaseF-digested MSLN from the above samples led to the identification of fourteen peptides in rMSLN (52.3% coverage) and nine peptides in the cancer cell lines, ten in the case of Capan-2, with a coverage of 30.7% and 35.3%, respectively (Table 5).

### 3.3. Characterization of Mesothelin N-Glycans

#### 3.3.1. *N*-Glycan Sequencing by HILIC-UPLC

MSLN *N*-glycan structures for the selected cell lines and rMSLN were characterized by *N*-glycan sequencing. It consists in the release and purification of *N*-glycans, which are then fluorescently labelled. These are separated in a hydrophilic column, with the retention time being higher as the *N*-glycans are more polar. The purified *N*-glycans can also be digested prior to the chromatography analysis with linkage and saccharide-specific exoglycosidases. The decrease in the retention time of a peak (measured in glucose units (GU) obtained from standardization with a dextran ladder) after an exoglycosidase digestion indicates how many of the specific, digested monosaccharides were present in that glycan structure, and the subsequent treatments with the corresponding specific exoglycosidases digest the structure until the basic *N*-glycan core. Finally, the stacking of all of the obtained chromatograms and the analysis of the GU shifts let the reconstruction of the glycan structures, being the areas under each peak indicative of the abundance of that particular *N*-glycan. In addition, the obtained GUs were compared with our GU database for *N*-glycans, Glycostore (https://glycostore.org, accessed on 22 April 2022) [58,59], to confirm the assigned structures.

The MSLN purification was required for that purpose. Thus, the MSLN from the conditioned media was immunopurified, using a monoclonal antibody coupled to protein G magnetic beads, as detailed in the methods section (Section 2.4). This purification was verified through SDS-PAGE followed by WB and Coomassie staining, as well as through the peptide mapping of the corresponding Coomassie bands. The WB of the immunopurified and unbound fractions demonstrated that MSLN was almost completely captured with this procedure and the Coomassie staining of the immunopurified fraction showed a solid single band corresponding to MSLN (Appendix A). The peptide analysis after tryptic digestion of the observed Coomassie bands also allowed the identification of MSLN, with a sequence coverage of around 33% (30.7–35.3%, depending on cell line) (Table 5; Appendix A, Appendix A).

The *N*-glycan sequencing clearly showed different structures between the MSLN purified from human cell lines and rMSLN, produced in murine cells. Throughout the results, the structures’ abundance is represented by their percentage among all of the assigned structures. A schematic representation for all of the assigned structures, their retention time and abundance in the indicated profile is shown in Appendix A (Appendix A).

The complete identification of the rMSLN *N*-glycans through the exoglycosidase digestions showed that they are composed of complex core fucosylated (α1,6 linkage) tri-antennary structures with terminal α-galactose moieties, differing in the number and position of the terminal alpha galactose and the presence or absence of external fucoses and sialic acids (Figure 2). SA was found on 42.2% of the structures, with a predominance of Neu5Gc (31.2%) over Neu5Ac (11.0%). Neu5Gc presents an extra -OH group to Neu5Ac, rendering it more polar. The digestion of Neu5Gc produces a higher GU decrease, which enables its differential assignment. Regarding their linkage, α2,3-SA were overexpressed against α2,6-linked SA (23.3% vs. 15.3%). SA α2,3 or α2,6-linked were assigned by differential shift after the NAN1 or ABS digestion. A single disialylated structure carried both α2,3 and α2,6 SA (3.6%). We also observed several *N*-glycans with external fucosylation on galactose (27.1%), as well as a structure carrying a *N*-acetyllactosamine (LacNAc) repeat (4.0%). The identification of the *N*-glycan isomers, consisting of tetra-antennary or tri-antennary structures containing LacNAc repeats, was performed after BTG digestion, as those structures with the LacNAc motif have one less terminal galactose to be digested.

Regarding the human MSLN from the cells-conditioned media, most of the structures corresponded to the complex core fucosylated-*N*-glycans with terminal sialylation. The main glycans were assigned from the identification obtained after specific exoglycosidases digestion. The conditioned media from Ovcar-8 cells yielded a higher amount of MSLN *N*-glycans than the PaC cells, and this allowed the performance of a complete panel of exoglycosidase digestions to characterize their glycans (Figure 3). The undigested profile showed a high *N*-glycan heterogeneity, mainly due to the differences in sialylation as the sialidase treatment gave rise to a more reduced number of complex *N*-glycans. Actually, sialylation confers such a massive heterogeneity that *N*-glycan structures could be poorly assigned in the undigested profile. The SA was mainly found in α2,3 linkage, as the digested profile for NAN1 (α2,3-SA digestion) only differs in a single peak (GU = 11.16, 18.3%) from ABS digestion (both α2,3-SA and α2,6-SA digestion). The major structure after ABS digestion was a core fucosylated, tetra-antennary structure (23.1%), together with its bisected form (8.0%) (GU = 10.11, +ABS digestion). In addition, the core fucosylated, tetra-antennary *N*-glycans with a LacNAc repeat were identified (15.1%). In total, the tetra-antennary *N*-glycans accounted for about 57.8% of all of the glycans, as other non-core fucosylated (11.6%) structures were present. The complex bi- and tri-antennary structures were less prevalent, accounting for about 13.5% and 28.7%, respectively. Approximately two thirds of these tri-antennary glycans were carrying a LacNAc repeat. The bisecting GlcNAc determinant was also present, in about the 28% of all of the glycans. Core fucosylation was found in most of the glycans (about 86%), which digested to the unfucosylated ones after BKF digestion. Some of the glycans (19.5%) also showed outer arm fucosylation, as shown by the digestion with AMF.

The MSLN *N*-glycans from PaC cells were analyzed after ABS digestion (Figure 4) because terminal sialylation caused such heterogeneity in the undigested profile that most of the structures could not be distinguished from the background in that profile (data not shown), thus pointing out the vast extension of sialylation. The major neutral structures obtained after ABS digestion were assigned from their GUs in Glycostore and through comparison with the structures obtained in the ABS digestion of the Ovcar-8 cells. We observed several complex fucosylated *N*-glycan structures assigned to bi-, tri- and tetra-antennary glycans, but no single one was prevalent as the core fucosylated, tetra-antennary structure found in the Ovcar-8 cells. Capan-2 and AsPC-1 MSLN shared similar *N*-glycans, which highly differed from those on HPAF-II. The latter only displayed three main structures which could not be assigned, and so the results’ interpretation is focused on the former. In Capan-2 and AsPC-1, only the complex core fucosylated structures were identified: approximately 8.5% bi-antennary (10.5/6.7); 21.5% tri-antennary (21.8/21.1) and 70% tetra-antennary (67.7/72.1) *N*-glycans. Among these tetra-antennary structures, 23% (15.0/31.0) carry a LacNAc repeat and 21% (27.1/14.5) carry two repetitions, while the remaining 26% (25.7/26.7) are not elongated by these repetitions. The low levels of *N*-glycans precluded further exoglycosidase digestions, and therefore the presence of outer arm fucosylation could not be assessed on the PaC samples.

#### 3.3.2. Glycan Determinants’ Expression by WB with Lectins and Sialidase Digestion

The characterization of the MSLN *N*-glycans shows the main structures present on MSLN, but it is not an option for a regular clinical PaC diagnosis assay based on specific MSLN glycoforms, as not enough quantity of purified MSLN from serum or tissue could be obtained for its glycan sequencing analyses. Thus, we focused on the analysis of the MSLN glycan determinants previously described by *N*-glycan sequencing, in particular sialylation, fucosylation, antenna branching and bisecting GlcNAc, using specific lectins. The selected lectins were: SNA which binds to α2,6-SA; MAL-II which recognizes most α2,3-SA glycans; AAL that binds to fucose determinants; PhoSL specific for core fucosylation; PHA-L that recognizes β1,6-antenna on tri- and tetra-antennary structures and PHA-E that binds to bisecting GlcNAc structures. We also used the PNA lectin, which recognizes Galβ3GalNAc, including the T antigen.

We analyzed these glycosylation determinants on immunopurified MSLN through WB. Equal amounts of total protein were immunopurified, using specific antibodies against MSLN bounded to protein G-agarose beads. Then, the protein immune complexes were separated by SDS-PAGE, blotted, and detected with specific lectins. Stripping and reblotting for MSLN was performed with all of the membranes, thus verifying the co-localization for both the glycans and protein. The relative abundance for each MSLN glycan-determinant band “normalized” against the MSLN protein band after stripping represented the amount of carbohydrate determinant per unit of MSLN. Thus, we could observe to what extent the carbohydrate determinants were expressed on the cell lines’ MSLN, as well as on rMSLN.

Sialylation was not detected by SNA or MAL-II lectins in any of the studied samples using this methodology (Figure 5a,b), except for the detection of α2,6-SA in rMSLN when loading the sample under reducing conditions. In this particular case, the quantity of the MSLN detected after reblotting the membrane with anti-MSLN antibodies was higher than in the cell lines, suggesting that the detection of the sialylated MSLN with lectins required a higher amount of the immunopurified protein. Taking into account that the *N*-glycan sequencing had showed SA on MSLN glycans in all of the samples, we decided to confirm the SA presence through specific sialidase digestions with NAN1 and ABS, followed by WB with anti-MSLN antibodies (Figure 5h). The results showed that in all of the cell lines, except HPAF-II, the digestion of α2,3-SA by NAN1 decreased the MSLN molecular weight as much as the total release of SA with ABS digestion, thus indicating that MSLN from all of the samples except HPAF-II was sialylated, and that sialylation was mostly α2,3-linked, as has been described for Ovcar-8 in the *N*-glycan characterization. For HPAF-II, only the neutral structures were present, as no change was detected after the sialidase digestions.

On the other hand, we were able to detect fucosylation, branching and bisecting GlcNAc (Figure 5c–f). The core fucosylation was detected on all of the cell lines using PhoSL (specific for core fucosylation), and regarding overall fucosylation (detected by AAL) all of the cells were positive, except AsPC-1. Antenna branching was assessed through the detection with PHA-L, which can detect the β1-6 ramification in branched *N*-glycans (Galβ4GlcNAcβ6(GlcNAcβ2Manα3)Manα3). This feature was shown in all of the cell lines, and was especially remarkable in AsPC-1. Bisecting the GlcNAc detection with PHA-E was also found on MSLN from all of the cell lines to a different degree. Lastly, the presence of terminal galactoses with the PNA lectin, which preferentially binds to Galβ3GalNAc, could not be detected (Figure 5g). Altogether, PhoSL, PHA-L and PHA-E could be useful for developing a specific assay to detect core fucose, branching and bisecting GlcNAc on MSLN.

### 3.4. Mesothelin Expression in Pancreatic Tissues

After the MSLN *N*-glycan characterization in the cell lines, we focused on analyzing the presence of the specific MSLN glycoforms in PaC tissues. The MSLN expression was assessed by WB on tissue lysates corresponding to the different staging of the PaC patients and the control individuals (Figure 6). MSLN was detected in 77.4% (*n* = 31) of the PaC tissues and in 10% of the control tissues (*n* = 10), which corresponded to the non-tumor tissue adjacent to the pancreatic tumor. The MSLN was also slightly expressed in two tissues from other gastrointestinal malignancies. The MSLN bands in the PaC samples were observed in 5/6 (83.3%), 16/21 (76.2%), 2/2 (100%) and 1/2 (50%) of the stages IIA, IIB, III and IV patients, respectively.

### 3.5. Core Fucosylated Mesothelin Abundance in Pancreatic Tissues

Once the presence of MSLN in the pancreatic tissues was analyzed, we then moved to determine their specific glycoforms in those tissue samples that showed a higher MSLN expression. As WB is just a semiquantitative methodology, we aimed to develop a combined ELISA/ELLA assay to quantify the specific MSLN glycoforms; the former was to measure the MSLN concentration, while the latter was to determine the specific glycan moieties levels on MSLN, using specific lectins against particular glycans. The ratio between the MSLN glycans and MSLN protein concentration would indicate the abundance in which the particular glycans are found on MSLN. Considering the previous results using lectins, we aimed to quantify fucosylation, branching and bisecting GlcNAc expression using the lectins PhoSL, PHA-L and PHA-E, respectively. The preliminary assays using PHA-E and PHA-L did not provide enough sensitivity to detect MSLN glycoforms. Therefore, we established an assay using PhoSL to determine the core fucosylated-MSLN (Cf-MSLN).

We developed a sandwich ELISA assay for the quantification of the MSLN and an ELLA assay to specifically evaluate the Cf-MSLN. An 8-point calibration curve (0–10 ng/mL) was used to quantify the MSLN. The LOD was 0.11 ng/mL, and linearity was demonstrated across 0.25–5 ng/mL. Intra- and inter-assay variation were of <5% and <17%, respectively. For the detection with PhoSL, a different 8-point calibration curve was used (0–40 ng/mL). In this case, the LOD was 0.41 ng/mL and the lineal range 1–10 ng/mL. The intra- and inter-assay variations were of <4% and <19%, respectively.

We used this ELISA/ELLA methodology to quantify the Cf-MSLN in eight tissue samples (two IIA (patients 3, 6), six IIB (patients 2, 4, 7, 16, 17, 20)) with high MSLN expression by WB. As tissue lysates comprise a rather complex biological matrix, we were required to perform a pre-purification step of the samples to reduce the background. We proceeded with MSLN immunopurification, using an antibody covalently bounded to magnetic beads, and so avoiding the elution of the antibody together with the sample. The purity and recovery of the immunopurification process were verified by WB and silver staining (Figure 7a).

The core fucosylated-MSLN glycoforms from PaC tissues were quantified and relativized to MSLN levels. The ratio of Cf-MSLN/MSLN was lower than the ratio obtained from the recombinant standard MSLN (0.66 ± 0.26 and 0.41 ± 0.11 for stage IIA and IIB patients, respectively). As the healthy pancreatic tissues do not express MSLN, a comparison with a healthy group was not possible. Our results also suggested that Cf-MSLN abundance in PaC tissues could decrease along the disease progression, as levels in the stage IIB patients were inferior to those in the stage IIA individuals.

### 3.6. Serum Core Fucosylated Mesothelin as PaC Biomarker

The core fucosylated-MSLN levels in pancreatic cancer tissues could scarcely became a routinely used biomarker, due to the invasiveness of the sample obtention. As mesothelin is shed from the cell membrane, we next analyzed the Cf-MSLN abundance in serum samples from 15 PaC patients, and compared it with the levels of 5 healthy donors and 11 ChP patients.

Protein concentration and matrix complexity is much higher in blood serum than in tissue lysates. For this reason, the pre-purification step proposed for the latter was unable to avoid the background in the serum samples (Figure 7b, IP1). Thus, these samples needed a double pre-purification process, consisting of a second immunopurification of MSLN using the same antibody magnetic beads. As a result, the second immunopurification reduced the background to accurately quantify the Cf-MSLN (Figure 7b, IP2).

The pancreatic cancer patients’ Cf-MSLN abundance was significantly decreased with respect of the control groups (from 0.93 ± 0.13 to 0.74 ± 0.20) (Figure 8a). When dividing the control group into healthy individuals and ChP patients, statistical differences were observed between the latter and the PaC patients, while they were not observed among the healthy and PaC patients, due to the lack of statistical power because of the low numbers of healthy individuals (*p* = 0.053). A ROC curve was calculated to determine the diagnostic potential of Cf-MSLN, which can discriminate the PaC patients from the rest of the control individuals with an area under the curve (AUC) of 0.806 and a sensitivity and specificity of 73.3% and 75.0%, respectively, with a cut-off value of 0.84 (Figure 8b).

The Cf-MSLN performance was compared in the studied cohort to that of mesothelin protein levels. These were quantified with the commercial ELISA kit Mesomark^®^, which recognizes soluble MSLN-related peptides. In this case, no significant differences were observed between the healthy, ChP and PaC individuals (Figure 8c,d), and, as described, the MSLN levels were not useful for PaC diagnosis.

## 4. Discussion

MSLN is a glycoprotein de novo expressed in a number of cancers, including PaC [41], which makes it attractive as a diagnostic and therapeutic target [33]. Despite the fact that its role in the malignant transformation is not fully understood, several authors have proved its influence in the invasive and metastatic potential, growth and apoptosis regulation and epithelial-to-mesenchymal transition [34].

Due to its limited expression in healthy tissues, MSLN has grown into a therapeutic target for different types of cancers and therapeutic strategies, such as vaccines [60], immunotoxins [61,62,63,64], monoclonal antibodies [65,66], antibody-drug conjugates [67,68] or CAR-T cells [69]. On the other hand, MSLN has also been proposed as a biomarker for various malignancies. Actually, there is an FDA-approved test for the diagnosis of malignant pleural mesothelioma based on soluble MSLN-related peptides’ serum levels [39,51,70]. Nonetheless, it has not been useful for the diagnosis of patients with other cancers, including PaC [50,71], as we also show in the cohort of patients analyzed in this work.

The expression of MSLN and the analysis of its glycosylation from PaC cells and tissues was addressed in this study, to determine the main glycan determinants present in MSLN. Even though the MSLN *N*-glycan presence had already been reported [54], these *N*-glycans were not characterized -to date. It is important to define the particular MSLN glycoforms in cancer, because a specific protein glycoform could improve the performance of the protein as a cancer biomarker. This is the case of the α2,3-SA PSA glycoforms for prostate cancer [72], core fucosylated-AFP glycoforms (AFP-L3) for hepatocellular carcinoma [73] or CA125-STn for ovarian cancer [74]. In addition, this knowledge could be useful for therapies targeting MSLN using specific antibodies [28,41], because their affinity could be dependent on the type of MSLN glycans, as it has been described for other glycoproteins [75,76].

In this work, we showed that the PaC cell lines express MSLN, which is in agreement with other reports [65,77]. In addition, the MSLN expression was also analyzed in PaC tissue lysates, which were mainly positive by WB (77%). This percentage is in accordance with other studies that report high expression levels in PaC (60–95%), while the presence in normal pancreatic tissues or other controls was scarce [42,43,44,45,46,47,78,79]. 

MSLN digestion with PNGaseF from PaC cells confirmed the existence of *N*-glycans, as previously described [54]. The molecular weight difference from the glycosylated to the PNGaseF-digested MSLN, observed by SDS-PAGE and MS, suggested a full occupancy of the three *N*-glycosylation sites and was confirmed by peptide mapping after the PNGaseF digestion.

The *N*-glycans of immunopurified MSLN from the PaC cell lines were characterized by *N*-glycan sequencing and by WB with several lectins, proved to be complex-type sialylated, most of which were highly branched and with core fucosylation. The α2,3-SA was predominant on the MSLN glycans for all of the cell lines, except for HPAF-II, which only showed neutral *N*-glycan structures. These glycan determinants coincided with the general pattern found in the PaC glycoproteome, which was extensively reviewed [16,24,80,81,82,83]. The predominance of α2,3-SA in the PaC cell lines is in agreement with previous studies [21] and contributes to the immunosuppressive properties of PaC [84].

The *N*-glycans detected in the commercial recombinant standard, produced in NS0-derived cells, were different from those in the human cells-MSLN, with the most remarkable features being the terminal α-galactose and the presence of Neu5Gc. The murine myeloma cell lines, which are often used as a production platform for the recombinant glycoproteins, yield high sialylation predominantly in α2,6 linkage, as well as considerably high levels of terminal alpha galactose and Neu5Gc [85]. The recombinant MSLN glycosylation mostly agrees with all of these statements, except for the SA linkage, which was mainly α2,3-SA.

In order to analyze the MSLN glycoforms from the PaC tissues and serum, a quantitative methodology based on antibody-lectin assays was developed, aiming to analyze the core fucosylated-MSLN (Cf-MSLN) previously identified in the PaC cells. Cf-MSLN levels were lower in the PCa tissues than the full core fucosylated-rMSLN. When analyzing the ratio of Cf-MSLN/MSLN in the serum samples from the PaC patients, other pancreatic disorders and the healthy volunteers, a significant decrease in the Cf-MSLN glycoforms was observed in the malignant stages, which is in agreement with the low Cf-MSLN levels described in PaC tissues.

Other authors have also described changes in the serum level of the core fucosylated-glycoforms from different proteins in PaC patients. This is the case for haptoglobin, which is mainly produced by the liver. Haptoglobin core fucosylation-levels were increased in the PaC patients compared to the healthy volunteers [25], but these levels were also raised in ChP, which limits its usefulness as a PaC biomarker. Another example was reported for serum ribonuclease 1, which is mainly from an endothelial origin. Serum ribonuclease 1 showed increased core fucosylated-glycoforms compared to the healthy controls [24]. In addition, an increase in core fucosylation has been described in the PaC serum glycome [22], mainly in the tri- and tetra-antennary structures. These authors analyzed the general glycosylation changes from the whole serum glycoproteins, released from different tissue sources, which do not necessarily reflect the specific glycosylation pattern from th ePaC glycoproteins.

The Cf-MSLN has the potential to discriminate the PaC from the control patients, while, in the same cohort, the protein MSLN levels do not have this ability, as reported by other authors [50,53]. These results emphasize the need for further studies evaluating Cf-MSLN in a larger cohort of patients, and including sera from other types of cancers in order to evaluate its specificity for PaC. Altogether, they show the importance of focusing on specific protein glycoforms altered in cancer, as well as exploring new glycoproteomic strategies [86,87] to high-throughput the analysis of Cf-MSLN and validate it as a PaC biomarker.

## 5. Conclusions

MSLN *N*-glycosylation has been characterized in PaC for the first time, and it consists mainly of core fucosylated, complex *N*-glycans, most of them α 2,3-sialylated. Moreover, serum MSLN core fucosylated-glycoforms were quantified by a sandwich ELLA, and showed a significant decrease in the PaC patients versus the control patients (healthy individuals and ChP patients), which paves the way of them to be used as PaC biomarkers.

## Figures and Tables

**Figure 1 biomedicines-10-01942-f001:**
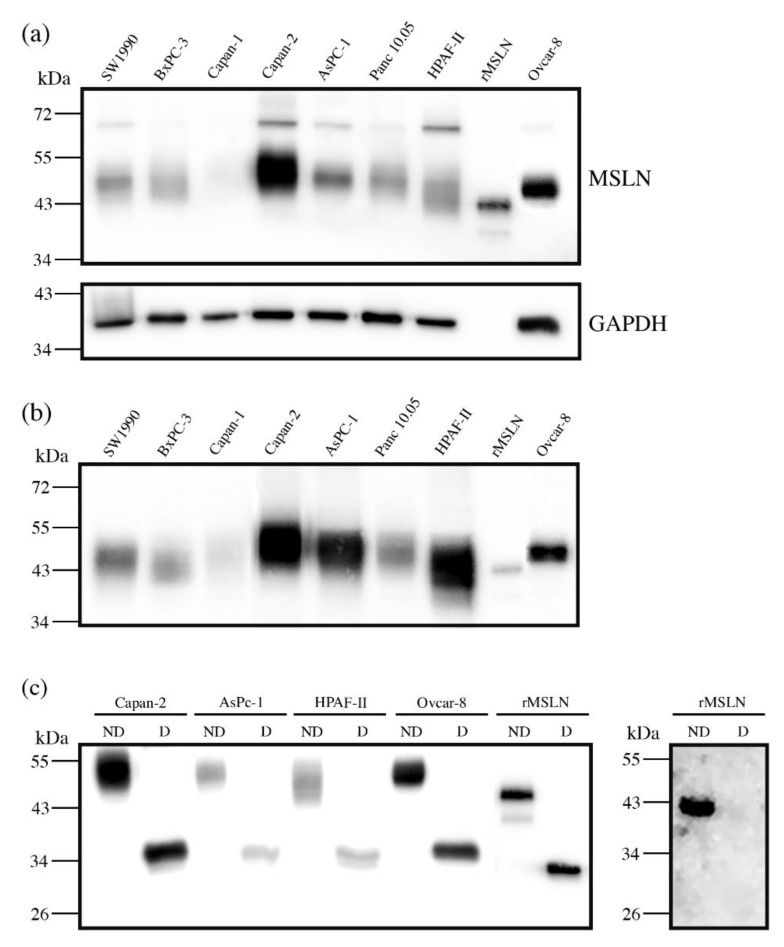
Mesothelin expression in PaC and ovarian cancer cell lines by WB (**a**,**b**), and with or without PNGaseF digestion (**c**). (**a**) MSLN expression in cell lysates (top), stripping and reblotting with GADPH as loading control (bottom): 20 µg total protein loaded per lane; (**b**) MSLN expression in secretion media: 25 µg total protein loaded per lane. For (**a**,**b**) 25 ng rMSLN were used as a positive control; (**c**) MSLN from 10 µg protein lysates or 25 ng rMSLN detected with an anti-MSLN antibody (clone MN-1) under reducing conditions (left) and WB on 50 ng rMSLN detected with *Aleuria aurantia* lectin (fucose detection) under non-reducing conditions (right); ND: not digested; D: digested.

**Figure 2 biomedicines-10-01942-f002:**
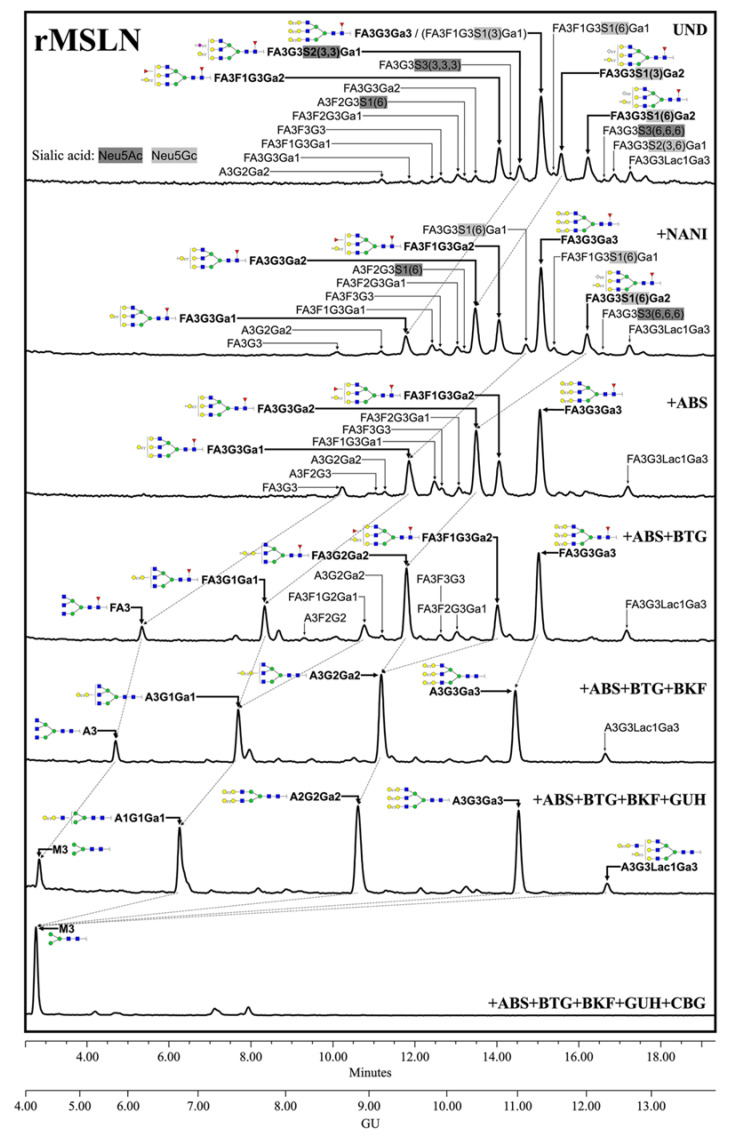
HILIC-UPLC profiling of rMSLN *N*-glycans labelled with 2-AB. From top to bottom, chromatograms for the undigested profile (UND) and after digestion with the specified exoglycosidases. Retention times were standardized against a dextran hydrolysate with glucose units (GU). Nomenclature used: F, at the start of the abbreviation indicates a core fucose α1,6-linked to the inner GlcNAc; Ax, the number (x) of antenna (GlcNAc) on tri-mannosyl core; A1, mono-antennary; A2, biantennary with both GlcNAcs β1,2-linked; A3, tri-antennary with two GlcNAc linked β1,2 to both mannoses and the third GlcNAc linked β1,4 to the α1,3 linked mannose; A4, tetra-antennary with GlcNAcs linked as in A3, and with an additional GlcNAc β1,6 linked to α1,6 mannose; Gx, number (x) of β1,4 linked galactose on antenna; Lacx, number (x) of *N*-acetyllactosamine repeats; Gax, number (x) of terminal α1,3 galactoses; Fx, number (x) of fucoses linked α1,3 to antenna GlcNAc; Sx, number (x) of sialic acids linked to galactose and (3),(6) indicates the sialic acid linkage α2,3 or α2,6, respectively. Most abundant *N*-glycan structures are shown and represented following symbol nomenclature for glycans (SNFG) guidelines. Dotted arrows represent the mobility of main peaks after subsequent exoglycosidases digestions. All glycans with their corresponding GUs are depicted in Appendix A (Appendix A).

**Figure 3 biomedicines-10-01942-f003:**
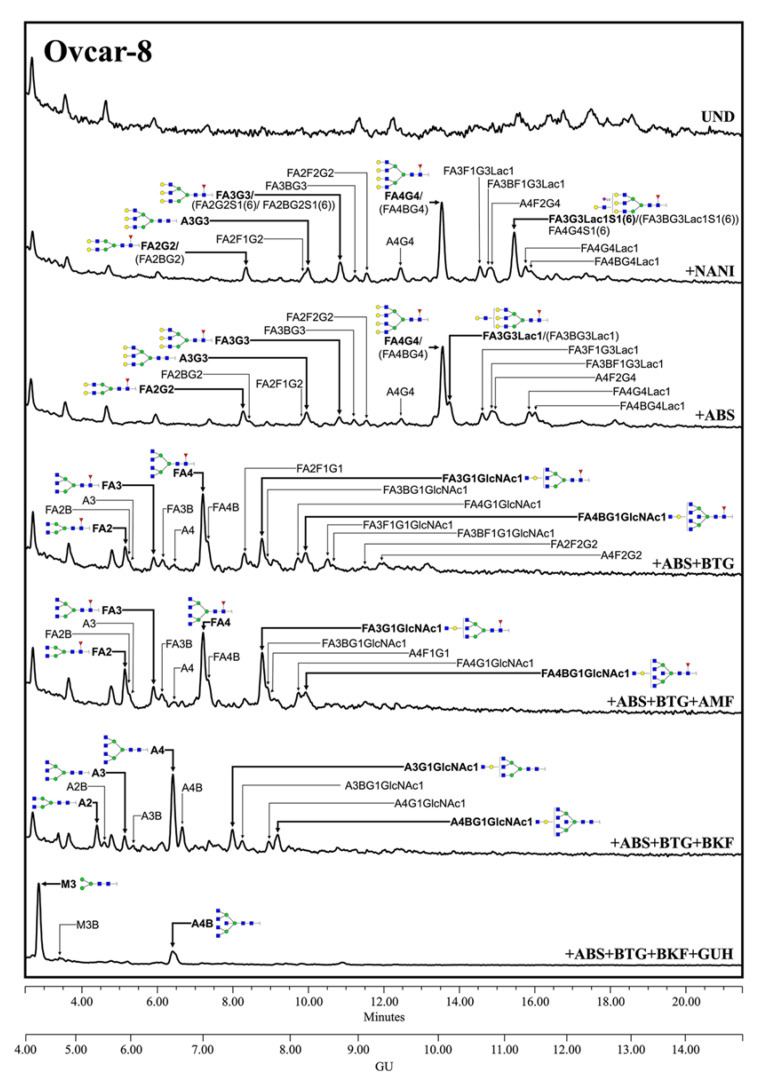
HILIC-UPLC profiling of MSLN *N*-glycans labelled with 2-AB from Ovcar-8 conditioned media. From top to bottom, chromatograms for the undigested profile (UND) and after digestion with the specified exoglycosidases. Retention times were standardized against a dextran hydrolysate with glucose units (GU). Nomenclature and representation of glycans are described in Figure 2. B indicates bisecting GlcNAc linked β1,4 to the β-mannose of the *N*-linked glycan core. All glycans with their corresponding GUs are depicted in Appendix A (Appendix A).

**Figure 4 biomedicines-10-01942-f004:**
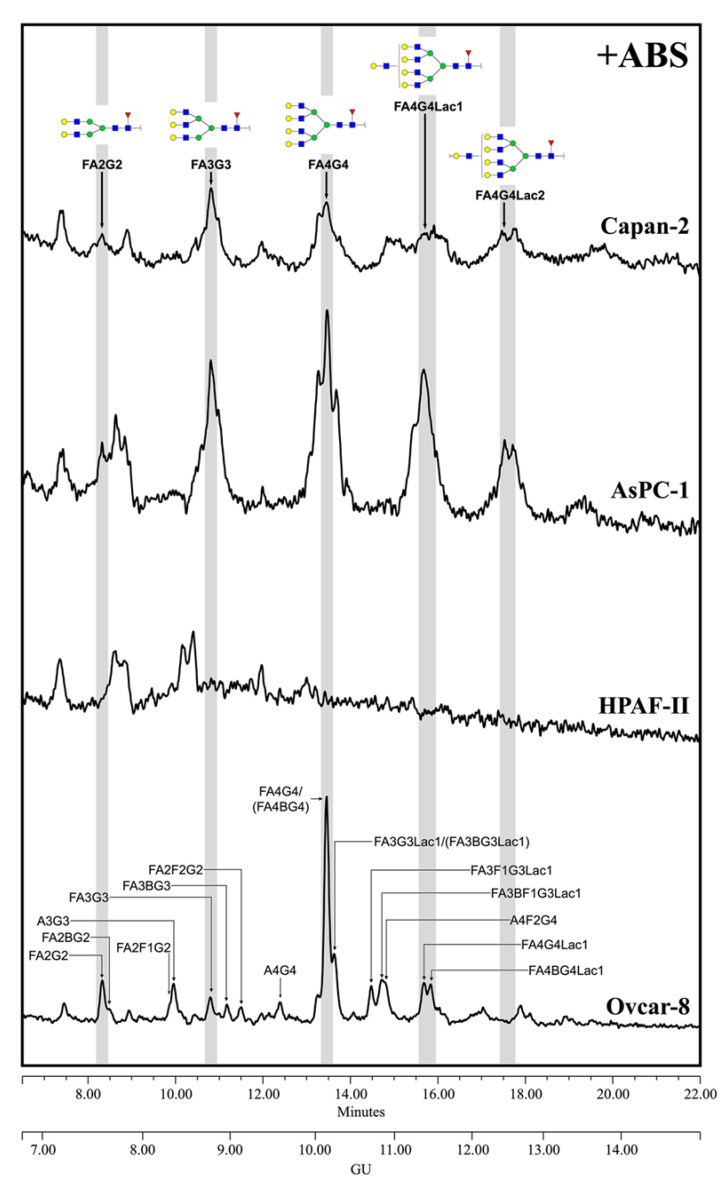
HILIC-UPLC profiling of MSLN *N*-glycans labelled with 2-AB from the different cell lines after ABS digestion. Retention times were standardized against a dextran hydrolysate with glucose units (GU). Nomenclature and representation of glycans are described in Figure 3. All glycans with their corresponding GUs are depicted in Appendix A (Appendix A).

**Figure 5 biomedicines-10-01942-f005:**
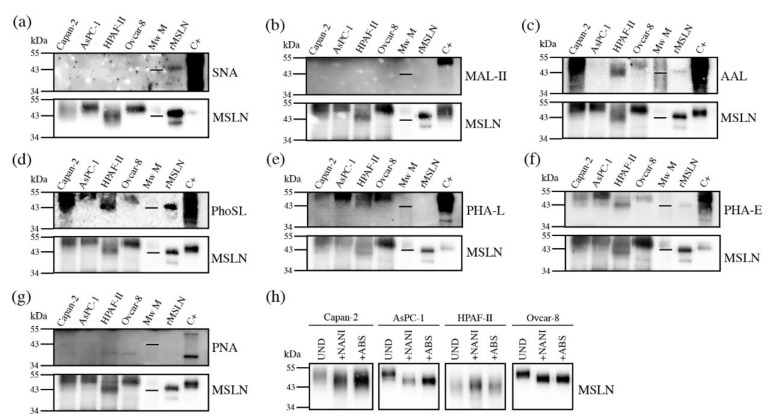
Detection of glycan determinants on immunopurified MSLN from cell lines conditioned media by WB with lectins (**a**–**g**), and WB of MSLN from conditioned media treated with sialidases (**h**). Samples loaded under non-reducing conditions except for panel (**a**). For each lectin, top panel shows the specific glycan determinant recognition, while bottom panel shows the result after membrane stripping with MSLN detection. Mw M: molecular weight marker -with the 43kDa line represented-; C+: protein lysate used as positive control for glycan-lectin recognition. (**a**) SNA for α2,6-SA; (**b**) MAL-II for α2,3-SA; (**c**) AAL for fucoses; (**d**) PhoSL for core fucose; (**e**) PHA-L for β1,6-antenna; (**f**) PHA-E for bisecting GlcNAc; (**g**) PNA for T-antigen (Galβ3GalNAc); (**h**)WB of MSLN from conditioned media. UND: undigested; +NAN1: α2,3-SA digestion; +ABS: all SA digestion.

**Figure 6 biomedicines-10-01942-f006:**
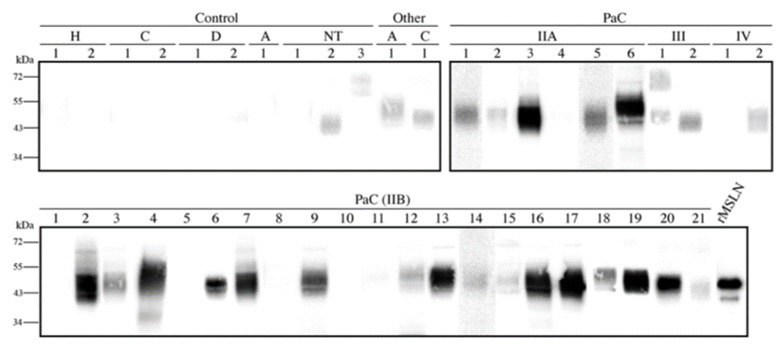
Mesothelin expression in pancreatic tissue lysates by WB under non-reducing conditions. Each number represents a different patient in each condition. A total of 20 µg protein was loaded per lane, and 25 ng rMSLN used as a positive control. All membranes were equally exposed to chemiluminescence for lane comparison. Control samples included two healthy pancreas (H) and pancreatic non-tumor tissues adjacent to the cancer region from cholangiocarcinomas (C), from duodenum adenocarcinomas (D), from ampulloma (A) and from PaC (NT). Cancer tissues included thirty-one PaC of different stages and other tumors: one ampulloma (A) and one cholangiocarcinoma (C)).

**Figure 7 biomedicines-10-01942-f007:**
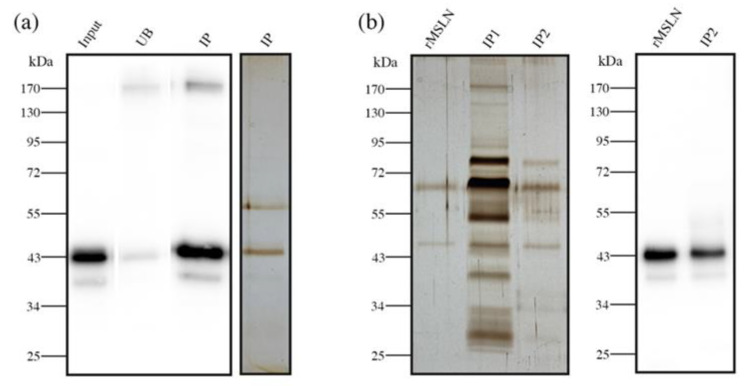
Sample pre-purification before Cf-MSLN quantification by ELISA/ELLA. (**a**) MSLN immunopurification from PaC tissue lysates. WB with anti-MSLN antibody of rMSLN spiked in tissue lysate (input), unbound (UB) and immunopurified (IP) fractions. Silver staining of the immunopurified fraction; (**b**) Silver staining of rMSLN spiked in serum after one (IP1) or two (IP2) successive immunopurification steps. WB with anti-MSLN antibody of the double immunopurified (IP2) fraction.

**Figure 8 biomedicines-10-01942-f008:**
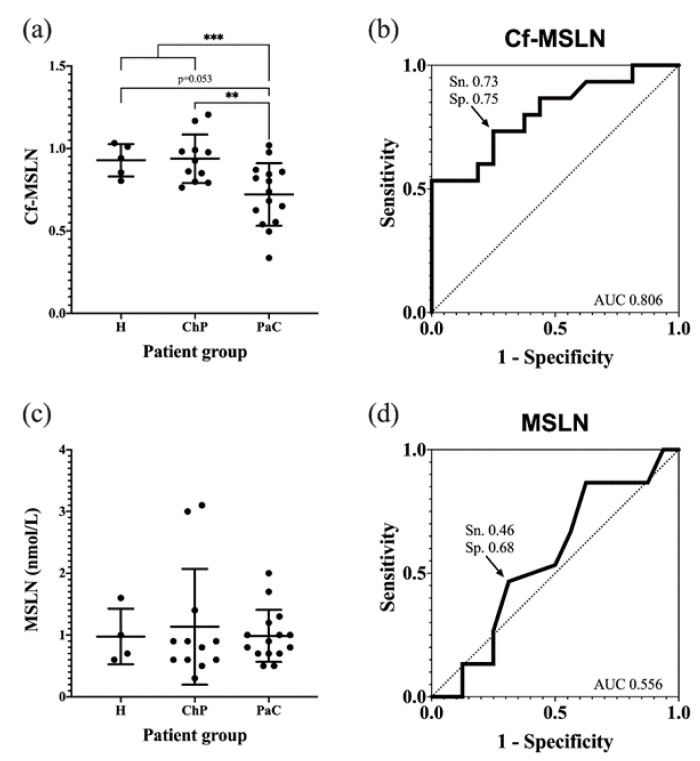
Quantification of Cf-MSLN and MSLN protein levels in serum samples from PaC patients and controls. (**a**) Cf-MSLN levels for healthy, ChP and PaC patients, represented as the mean ± SD for each group. ** *p* < 0.01, *** *p* < 0.001; (**b**) ROC curve for discriminating between PaC and control group using Cf-MSLN ratios; (**c**) MSLN protein levels for healthy, ChP and PaC patients, represented as the mean ± SD for each group. No significant differences were detected among groups; (**d**) ROC curve for discriminating between PaC and control group using MSLN protein levels.

**Table 1 biomedicines-10-01942-t001:** Frozen pancreatic tissue pieces for protein lysate obtention. Samples used for MSLN WB expression and/or Cf-MSLN quantification. Patients’ clinico-pathological characteristics.

Pathology	Cases	N	Male/Female	Age Average	Range
Control	Healthy	2	1/1	59	50–68
	NT cholangiocarcinoma	2	1/1	74.5	71–78
	NT duodenum adenocarcinoma	2	0/2	75	72–78
	NT Ampullary carcinoma	1	1/0	67	-
	NT PaC	3	2/1	68.3	64–77
PaC	IIA	6	4/2	69.5	64–78
	IIB	21	10/1	64.2	49–79
	III	2	2/0	58	52–64
	IV	2	2/0	70.5	65–76
Other gastrointestinal malignancies	Ampullary carcinoma	1	0/1	77	-
	Cholangiocarcinoma	1	1/0	78	-

**Table 2 biomedicines-10-01942-t002:** Serum samples for Cf-MSLN quantification. Clinico-pathological characteristics of individuals.

Pathology	Cases	N	Male/Female	Age Average	Range
Control	Healthy	5	3/2	59.4	44–75
	ChP	11	8/3	51.8	42–72
PaC	IIA	4	2/2	68.5	61–78
	IIB	3	2/1	73	64–81
	III	3	3/0	63	52–73
	IV	5	3/2	62	30–76

**Table 3 biomedicines-10-01942-t003:** Lectins used in WB.

Lectin	Glycan Determinant	Concentration	Reference
Biotinylated *Sambucus nigra Agglutinin* (SNA)	Neu5Acα6Gal/GalNAc	2 µg/mL	B-1305 (Vector Laboratories)
Biotinylated *Maackia amurensis* Lectin II (MAL-II)	Neu5Acα3Galβ3GalNAc	2 µg/mL	B-1265 (Vector Laboratories)
Biotinylated *Aleuria aurantia* Lectin (AAL)	Fucα3/6GlcNAc	2 µg/mL	B-1395 (Vector Laboratories)
Biotinylated *Pholiota squarrosa* Lectin (PhoSL)	Core fucose (α1,6)	2 µg/mL	[55]
Fluorescein labelled *Phaseolus vulgaris* Leucoagglutinin (PHA-L)	Galβ4GlcNAcβ6 (GlcNAcβ2Manα3) Manα3	2 µg/mL	FL-1111 (Vector Laboratories)
Fluorescein labelled *Phaseolus vulgaris* Erythroagglutinin (PHA-E)	Galβ4GlcNAcβ2Manα6 (GlcNAcβ4) (GlcNAcβ4Manα3) Manβ4	2 µg/mL	FL-1121 (Vector Laboratories)
Digoxigenin labelled Peanut Agglutinin (PNA)	Galβ3GalNAc	10 µg/mL	11 210 238 001 (Roche)

**Table 4 biomedicines-10-01942-t004:** MSLN tryptic peptides containing an *N*-glycosylation motif (N-X-S/T). Expected masses for the peptide if it was not glycosylated (N) or glycosylated (D) before PNGaseF digestion. Crossed boxes correspond to identified peptides on Capan-2 (C), AsPC-1 (A), HPAF-II (H), Ovcar-8 (O) and rMSLN (r) after PNGaseF digestion of the corresponding MSLN bands.

Peptide	Sequence	N Mass (Da)	D Mass (Da)	Identification									
				N					D				
				C	A	H	O	r	C	A	H	O	r
Peptide 92-101	W**NVT**SLETLK	1189.64	1190.62						X	X	X	X	X
Peptide 187-200	LAFQNM**NGS**EYFVK	1646.78	1647.76						X				X
Peptide 215-230	ALSQQ**NVS**MDLATFMK	1782.87	1783.85										X

**Table 5 biomedicines-10-01942-t005:** Mesothelin peptides detected by UPLC-ESI-QTof. Crossed boxes indicate peptides identified in Capan-2 (C), AsPC-1 (A), HPAF-II (H), Ovcar-8 (H), rMSLN (rM) and deglycosylated rMSLN (dM).

Peptide ID	Sequence	Retention Time (min)	Theoretical Mass (Da)	Experimental Mass (Da)	Mass Deviation	Identification					
						C	A	H	O	rM	dM
84-90	MSPEDIR	7.06	846.3905	846.4038 (424.2097 + 2H+)	+0.0133					X	X
102-108	ALLEVNK	8.94	785.4647	785.4818 (393.7487 + 2H+)	+0.0171	X	X	X	X	X	X
247-256	LLGPHVEGLK	12.11	1061.6233	1061.6517 (354.8917 + 3H+)	+0.0284	X	X	X	X	X	X
177-184	QLDVLYPK	15.16	974.5437	974.5490 (488.2823 + 2H+)	+0.0053	X	X	X	X	X	X
109-122	GHEMSPQVATLIDR	17.05	1552.7667	1552.7793 (518.6009 + 3H+)	+0.0126					X	X
266-270	DWILR	17.57	701.3860	701.4058 (351.7107 + 2H+)	+0.0198	X	X	X	X	X	X
201-214	IQSFLGGAPTEDLK	22.15	1474.7667	1474.7676 (738.3916 + 2H+)	+0.0009	X	X	X	X	X	X
233-246	TDAVLPLTVAEVQK	24.67	1482.8293	1482.8290 (742.4223 + 2H+)	−0.0003	X	X	X	X	X	X
187-200	LAFQNMNGSEYFVK	26.14	1646.7763	1647.7828 (824.8922 + 2H+)	+1.0065	X					X
92-101	WNVTSLETLK	27.01	1189.6343	1190.6230 (596.3193 + 2H+)	+0.9887	X	X	X	X		X
15-24	EIDESLIFYK	27.12	1255.6336	1255.6336 (628.8246 + 2H+)	±0.0000	X	X	X	X		
13-24 *	APEIDESLIFYK	28.40	1423.7235	1423.7188 (712.8672 + 2H+)	−0.0047					X	X
215-230	ALSQQNVSMDLATFMK	34.27	1782.8644	1783.8636 (595.6290 + 3H+)	+0.9992						X
44-58	VNAIPFTYEQLDVLK	35.96	1748.9348	1748.9320 (875.4738 + 2H+)	−0.0028	X	X	X	X	X	X
26-43	WELEACVDAALLATQMDR	42.51	2090.9765	2090.9950 (698.0062 + 3H+)	+0.0185					X	X
61-83	LDELYPQGYPESVIQHLGYLFLK	46.30	2721.4053	2721.3921 (908.1385 + 3H+)	−0.0132					X	

* rMSLN contains a residue swap in position 14 (arginine to proline) which removes a cleavage site for trypsin, thus giving rise to a different peptide than the one observed in cell lines.

## Data Availability

Data is contained within the article or Appendix A.

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
