# Peer review of "Characterization of Mesothelin Glycosylation in Pancreatic Cancer: Decreased Core Fucosylated Glycoforms in Pancreatic Cancer Patients’ Sera"

_biomedicines, 2022, doi:10.3390/biomedicines10081942_

Round 1
Reviewer 1 Report
The authors characterized pancreatic cancer neoexpressed mesothelin and have demonstrated the possibility of using MSLN glycosylation as a potential marker. The work is of great importance and well-deigned. The data and results adequately support the conclusion. I don’t have further comments on the manuscript.
Author Response
We thank the reviewer for the very positive comments about our study.
Reviewer 2 Report
In this research, Adrià Duran et al. reported the new finding that decreased core fucosylated glycoforms in pancreatic cancer patients’ sera, which may be used as PaC biomarkers. The authors characterized MSLN glycans from PaC cells and serum samples and analyzed their potential usefulness as PaC biomarkers. They developed an enzyme linked lectin assay to measure core fucosylated MSLN (Cf-MSLN) glycoforms. MSLN glycans in PaC cells were analyzed by glycan sequencing and western blotting with lectins. The Cf-MSLN glycoforms were quantified on PaC serum samples and Cf-MSLN was significantly decreased in PaC patients compared to control sera.
The paper is overall convincing. I think that the work is very interesting. The results and the implications will be of interest for cancer researchers and clinicians. The manuscript should be considered for publication, as long as the authors are able to address some specific concerns as follow.
1,Currently, the most used biomarker is the carbohydrate antigen 19-9 (CA19-9), corresponding to the sialyl-Lewis a antigen, primarily used in combination with CEA [4]. There are not full name of some words such as CEA , ChP, LOD and LOQ in the manuscript. There are some grammar problem such as a antigen in previous sentence.
2,This manuscript contains ten main figures that is quite long article structure. I would like to suggest putting some simple figures together, so it is easier for readers to read and find the information.
3,For the western blot, there is not protein control such as actin or GAPDH. I see the positive control as rMSLN, but usually we also provide actin or GAPDH as negative control for comparing the protein level in the cells.
4,In the figure1 b showed quite high expression in the HPAF-II supernatant, however, the cell protein level showed normal expression compare with cell lines such as AsPC-1 and Panc10.05. Is there any explain for this?
5,Some figures in the manuscript are fuzzy and indistinct and it is difficult to see the information in some figures.
6, The authors claimed that Regarding PaC stage, no correlation was observed between MSLN expression and disease progression. MSLN bands were observed in 5/6 (83.3%), 16/21 (76.2%), 2/2 (100%) and 1/2 (50%) of stages IIA, IIB, III and IV patients, respectively. When the authors made the conclusion no correlation between MSLN expression and disease progression, the number should be bigger in stage III and IV, because of the number of patients in stage III and IV only two.
7, The figure 10 c should show the line similar as a figure 10 a, which will clearly show which group comping which group.
8, As the discussion in the manuscript, MSLN has also been proposed as a biomarker for various malignancies, and how can we distinguish different tumors or other disease by MSLN?
9,What is the sensitivity of the MSLN as biomarker in serum if we apply this finding in clinic? Cf-MSLN is the specific PaC biomarker. Do you think it is enough Cf-MSLN testing in other type of cancer or disease to conclude specific PaC biomarker?
10, in the discussion part, there are several paragraphs, which only showed the similar results in the results part. It is better to revise this part. I would suggest some discussion about the problems and challenges of Cf-MSLN as biomarker in the research.
Author Response
We thank the reviewer for the overall comments on our study and expect that we have addressed all the raised issues (please see attached document).

Author Response
We thank the reviewer for the positive comments about our study and have addressed the minor issues indicated:
- What was the source of healthy pancreatic tissue (l.170)?
Tissues from two healthy pancreas came from autopsy. We have added that information in materials and methods section.
- L. 198 and 202 – abbreviations of sialidases sould be explained here, at their first use
We have included complete sialidases name at their first use in the materials and methods section.
- Fig. 6 – The blots are cut-off just at the band of interest. Could it be extended up to about 60 kDa?
Western blot analyses after MSLN immunopurification were cut-off at 55 KDa in order to avoid cross reactivity of the secondary antibody against the anti-MSLN antibody used in the purification protocol.
- L.609 – How do the Authors define over- and underexpression of the carbohydrate determinants, as actually there is no reference value to be compared to?
For general characterization of glycan determinants on MSLN from different cell lines, we immunopurified MSLN from equal amounts of total protein. Cell lines’ conditioned media were quantified using Bradford Assay (Bio-Rad) and afterwards MSLN from the same amount of total protein was purified in parallel using the same protocol. After lectin blotting, PVDF membranes were stripped and reblotted using MSLN antibody. Although western blotting methodology is a semiquantitative technique, comparison of lectin signal intensity normalized by the MSLN content allows to qualitatively determine which glycoepitopes are differently expressed among the cell lines as well as in rMSLN.
We have rephrased the sentence and now it reads: “we could observe to which extent carbohydrate determinants were expressed on the cell lines' MSLN, as well as on rMSLN”.
- The Authors are kindly asked to check the manuscript for spelling/typesetting errors. What I have noticed:
- l. 67 – acetylglucosamine
- l. 277 – with slight modifications or slightly modified
- l. 814 – significant decrease
English language and spelling have been revised along the manuscript.
Round 2
Reviewer 2 Report
The manuscript was well revised, and the reply answered my questions. So I think that it should be accepted in this revised manuscript.